# Influence of Blend Composition and Silica Nanoparticles on the Morphology and Gas Separation Performance of PU/PVA Blend Membranes

**DOI:** 10.3390/membranes9070082

**Published:** 2019-07-05

**Authors:** Hemmat Shirvani, Saeid Maghami, Ali Pournaghshband Isfahani, Morteza Sadeghi

**Affiliations:** 1Department of Chemical Engineering, Isfahan University of Technology, Isfahan 84156–83111, Iran; 2Institute for Integrated Cell-Material Sciences (iCeMS), Kyoto University, Kyoto 606–8501, Japan

**Keywords:** blend membranes, mixed-matrix membranes, polyurethanes, poly(vinyl alcohol), gas permeability

## Abstract

Polymer blending and mixed-matrix membranes are well-known modification techniques for tuning the gas separation properties of polymer membranes. Here, we studied the gas separation performance of mixed-matrix membranes (MMMs) based on the polyurethane/poly(vinyl alcohol) (PU/PVA) blend containing silica nanoparticles. Pure (CO_2_, CH_4_, N_2_, O_2_) and mixed-gas (CO_2_/N_2_ and CO_2_/CH_4_) permeability experiments were carried out at 10 bar and 35 °C. Poly(vinyl alcohol) (PVA) with a molecular weight of 200 kDa (PVA200) was blended with polyurethane (PU) to increase the CO_2_ solubility, while the addition of silica particles to the PU/PVA blend membranes augmented the CO_2_ separation performance. The SEM images of the membranes showed that the miscibility of the blend improved by increasing the PVA contents. The membrane containing 10 wt % of PVA200 (PU/PVA200–10) exhibited the highest CO_2_/N_2_~32.6 and CO_2_/CH_4_~9.5 selectivities among other blend compositions, which increased to 45.1 and 15.2 by incorporating 20 wt % nano-silica particles.

## 1. Introduction

Membrane technology plays an important role in reducing the manufacturing cost and energy consumption in industrial processes [1,2,3,4]. For many practical applications, such as CO_2_ removal from natural gas or CO_2_/N_2_ separation, seeking new materials with high gas permeability and selectivity is great importance [5,6]. Polymer blending and mixed-matrix membranes (MMMs) are attractive techniques for the development of membranes with high separation performance. In particular, polymer blending is a simple approach to combine the advantages of a highly permeable and a highly selective polymer pair to improve the properties of polymer membranes [7,8,9,10]. Besides, it has been shown that the incorporation of inorganic particles (e.g., silica, MOFs, zeolites) to a polymer matrix can improve the separation performance, mechanical properties, and control the aging and plasticization of the membranes [11,12,13,14].

The gas permeability of polymer blends mostly depends on the phase behavior. In general, the polymer blend morphology can be categorized into two types: (i) miscible and (ii) immiscible or partially miscible. In the miscible blend, the polymer pairs are entirely dissolved into each other and make a homogenous single-phase mixture. By contrast, the polymer pairs in immiscible blends are not dissolved into each other and make a separated phase by a weak polymer–polymer interface, which leads to inferior mechanical–thermal properties [15,16,17,18,19]. In recent years, partially miscible blends have been extensively used for gas separation applications, and their performance is dependent on the morphology, composition, and testing conditions (i.e., temperature, pressure) [20]. In mixed-matrix membranes, the selection of the filler and polymer matrix is important for the separation performance. The poor adhesion between the inorganic fillers and glassy polymers is a large challenge, which leads to non-selective voids and deteriorates the performance of the membrane [21]. Therefore, the blending of glassy and rubbery polymers has been proposed to overcome this problem. The high chain mobility of rubbery polymers provides strong interaction with inorganic fillers, which is a requirement for the fabrication of defect-free gas separation membranes [20,22].

Rubbery polymers are well-known materials for the development of membranes with high CO_2_/light gas selectivity. Moreover, increasing the CO_2_ solubility is highly demanding for the improvement of the separation performance, especially for CO_2_/N_2_ industries, where the molecular sieving does not play a significant role. Among rubbery polymers, PUs have been studied for CO_2_/N_2_ separation in recent years because of their structural versatility and good filmability. Besides, the ethereal groups in PUs are suitable sites for CO_2_ sorption. PUs are block copolymers consisting of hard and soft segments of urethane and polyether or polyester, respectively, which are usually phase separated. The soft segment is responsible for the elasticity properties of PUs, while the physical and mechanical features are controlled by the urethane and urea linkages [23,24]. Although they have high permeability and a good filmability, the selectivity of the PU membrane is not so attractive. 

There have been many efforts to improve the selectivity of PUs by adding inorganic particles or blending with highly selective polymers [25,26,27]. Typically, the selectivity of the membranes is enhanced by blending PUs with highly selective glassy polymers such as poly(methyl methacrylate) (PMMA), polyetherimide (PEI), and poly(amide-imide) (PAI) [8,27]; however, the gas permeability decreases as expected due to the lower free volume of the glassy polymers. In another study, the small addition a third component (polyethylene oxide-polypropylene oxide-polyethylene oxide triblock copolymer) to the PU/polyvinyl acetate solution leads to better compatibility of the blend and a higher gas selectivity [28]. The performance of PU MMMs is strongly dependent on the surface chemistry and the size of the particles. The tendency of the inorganic filler to interact with the hard or soft segments of PU can alter the separation properties. For example, higher phase separation and gas permeability was reported by the incorporation of cyanuric chloride and ZnO particles into the PU membranes [29,30]. However, in the PU/silica system, the particles are more likely to interact with the soft domains of the PU, decreasing the phase separation and the gas permeability, although the CO_2_/N_2_ selectivity is greatly improved by more than 80% [31].

Poly(vinyl alcohol) (PVA) is a promising hydrophilic material for CO_2_ capture and contains many hydroxyl groups. The good filmability, excellent mechanical properties, and good oxidation resistance are other advantages of the polymer, which make it a good candidate for gas separation application [32]. However, its crystallinity and high swelling tendency may limit further usage [33]. On the other hand, PU membranes have high permeability and moderate selectivity, therefore, PU/PVA blending was considered to improve the separation properties. In our recent publication, the effect of composition and molecular weight of PVA on the properties of PU/PVA was studied [32]. The gas permeability of the blend membranes decreased by the molecular weight and PVA content while the selectivity increased. The CO_2_/CH_4_ and CO_2_/N_2_ selectivity of the membranes was enhanced by more than 35% and 27%, respectively. 

In the present study, a series of PU/PVA blends at different compositions was synthesized. The effect of nano-silica particles on the physical and gas separation properties of the blend membranes was explored. The enhanced selectivity of PU/silica MMMs in our recent publications inspired this research. It is supposed that the synergetic effect of silica particles and PVA can result in more enhancement in the separation properties of the PU membranes.

## 2. Materials and Methods 

### 2.1. Materials

Poly(tetramethylene glycol) (PTMG, *M*_w_: 2000 g/mol) was kindly provided by Arak Petrochemical Company (Arak, Iran) and dried under vacuum at 80 °C for 24 h before use. Isophorone diisocyanate (IPDI), dibutyltin dilaurate (DBTDL) as the catalyst, and butanediamine (BDA) and N,N-Dimethylacetamid (DMAc) as the solvents were obtained from Sigma–Aldrich ( Darmstadt, Germany). The chain extender (BDA) was dried over 4 Å molecular sieve before use. Poly(vinyl alcohol) with an average molecular weight of 200 kg/mol (PVA200) and a hydrolysis degree of 99 percent was purchased from Sigma–Aldrich. Tetraethyl orthosilicate (TEOS) and 3-Glycidylozxypropyl trimethoxysilane (GOTMS) were provided by Sigma–Aldrich.

### 2.2. Polymer Synthesis 

The PU was synthesized by a two-step bulk polymerization method, described elsewhere [23]. First, an excess amount of IPDI (3.3 g, 15 mmol) was added dropwise to 10 g PTMG (5 mmol) at 70 °C under N_2_ atmosphere, followed by the addition of three drops (approximately 0.15 mL) of DBTDL as the catalyst. After mechanical stirring for 2 h, the exact amount of BDA (0.88 g, 10 mmol) was added to the reaction to carry out the chain extension. Scheme 1 shows the PU synthesis steps. The synthesized PU was precipitated and washed with an EtOH/water mixture (50/50 vol %) to remove the residual monomers, and dried under vacuum at 80 °C for further use. The average molecular weight (*M*_w_) and polydispersity index of the PU was determined at around 65.1 KDa and 1.4, respectively, by gel permeation chromatography (GPC, Shimadzu, 800 series, Kyoto, Japan).

### 2.3. Silica Synthesis 

Silica nanoparticles were synthesized via the sol-gel method by hydrolysis of TEOS in ethanol with hydrochloric acid as the catalyst. In this method, 25 g of TEOS and 4 g of GOTMS were mixed in 30 mL of dried methanol at 70 °C for 1 h. Then, the mixture of 30 mL methanol, 7.5 g water, and 0.83 g hydrochloric acid was gradually added to the solution. Finally, TEOS was hydrolyzed by mixing at 80 °C for 1 h.

### 2.4. Membrane Fabrication 

The PU/PVA200 blend membranes were prepared at different PVA loadings as described in our previous work. To fabricate MMMs, 10 wt % PU and 1 wt % PVA solutions were prepared in DMAc. Then, the exact amount of silica sol-gel solution was mixed with the PVA solution. Finally, the PVA/silica solution was added to the PU, and the mixture was stirred–sonicated several times. The solution was cast into Teflon Petri dishes and the film formed by slow evaporation of the solvent at 60 °C overnight and vacuum dried at 80 °C for 24 h.

### 2.5. Characterization

Fourier transform infrared spectroscopy (Jasco FTIR 680 Plus, Tokyo, Japan) was performed in the wavenumber range of 4000–400 cm^−1^ at room temperature. Wide angle X-ray diffraction (WAXD, Rigaku RINT XRD, Tokyo, Japan) of the samples was carried out by monitoring the diffraction pattern at 2θ = 5–40° and a scanning rate of 5°/min. The thermal transition and crystallinity of the samples were determined by differential scanning calorimetry (DSC, Bruker DSC 3100SA, Karlsruhe, Germany) between −100 to 200 °C at a heating rate of 10°/min. The morphology of the membranes was monitored by scanning electron microscopy (SEM, Philips XL30, Eindhoven, The Netherlands). The SEM samples were prepared by freeze-fracturing in liquid nitrogen, followed by gold/palladium coating to prevent charging.

### 2.6. Gas Permeation Analysis 

The pure and mixed-gas permeability of N_2_, O_2_, CH_4_, and CO_2_ through PU/PVA200 blend membranes and PU/PVA200/silica MMMs were assessed using the constant pressure method at 10 bar and 35 °C. The gas permeability coefficient of the membranes was calculated using the following equation:(1)$$P=\frac{ql}{A\left(p_{f}-p_{p}\right)}$$
where is permeability in Barrer (1 Barrer = 10^−10^ cm^3^ (STP) cm cm^−2^·s^−1^·cmHg^−1^), is the flow rate of the penetrants (cm^3^ (STP)·s^−1^) through the membranes, l is the membrane thickness (cm), p_f_ and p_p_ are the respective absolute pressures (cmHg) at feed and permeate sides. In addition, A represents the effective membrane area (cm^2^). The ideal selectivity, α_A/B_, of membranes is calculated from the pure gas permeation coefficients as follows:(2)$$\alpha_{A/B}=\frac{P_{A}}{P_{B}}$$

The mixed gas permeability of the membranes was measured at 10 bar and 35 °C under the gas mixture of CO_2_/N_2_ and CO_2_/CH_4_ (50/50 vol %). The composition of the permeance flow was detected by the gas chromatography machine (GC-2014, Shimadzu). The separation factor was determined by Equation (3):(3)αi/j=yiyjxixj
where y and x are the mole fraction of each component in the permeate and feed flow, respectively. The mixed gas permeability was calculated as follow:(4)$$P=J_{i}\frac{l}{\Delta p}=10^{10}\frac{273.15}{76}\frac{\mathrm{Vl}}{\mathrm{AT}}\left(\frac{\mathrm{dp}}{\mathrm{dt}}\right)\frac{y_{i}}{x_{i}\Delta p}$$

## 3. Results and Discussion

### 3.1. Chemical and Physical Characterization

The FTIR spectra of the PU/PVA200 blends are shown in Figure 1. The completion of the PU reaction can be monitored by the disappearance of the NCO peak at 2250 cm^−1^. The peak at 1110 cm^−1^ corresponded to the anti-symmetric stretching vibrations of C–O–C. The carbonyl bonds can be observed at 1600–1800 cm^−1^, and the NH stretching vibrations appearred at 3300 cm^−1^ [32]. The study of the two carbonyl peaks was helpful to understand the interactions and hydrogen bonding in the PU. The NH groups of the urethane linkage can be hydrogen bonded with proton accepting oxygen in the urethane C=O groups in the hard segments and C–O–C bonds in the soft segments. The type and strength of each hydrogen bonding can be identified by the magnitude and the shift of carbonyl bands. The first peak appearred at a lower frequency, ~1640 cm^−1^, corresponded to the bonded carbonyl groups, and the peak at around 1720 cm^−1^ related to the free carbonyl groups [34,35].

The FTIR spectra of PVA200 showed –CH_2_ bending at 1443 cm^−1^. The acetate groups of the non-hydrolyzed part of the PVA were observed between 1715 and 1750 cm^−1^. The vibration region from 2908 to 2940 cm^−1^ was related to C–H stretching of alkyl groups. The wide peak at 3200–3500 cm^−1^ was attributed to the stretching O–H group, resulting from the intramolecular and intermolecular hydrogen bonds. As illustrated in Figure 1, the intensity of the bonded carbonyl peak decreased by the addition of PVA200 to the PU blend, while the free carbonyl peak became stronger. Furthermore, the intensity of the hydroxyl peak of the PVA200 significantly decreased, and the absorption peak of the NH group in the PU broadened. This observation indicates the more phase-mixed PU structure with the addition of PVA200. It seems that the tendency of PVA to interact with the carbonyl groups disrupted the intermolecular hydrogen bonding in the PU structure.

The FTIR spectra of the MMMs are shown in Figure 2. The peaks at 797 and 3400 cm^−1^ were related to the symmetric Si–O–Si stretching and the hydroxyl groups of the silica particles, respectively. The intensity of the bonded carbonyl group of the PU decreased while the free carbonyl increased with the addition of silica particles, indicating a more phase-mixed structure of the PU. It is supposed that the silica particles mostly disperse in the soft segments and interact with the ethereal linkage of the polyol. This result was observed in other PU/silica MMMs, elsewhere [31,36].

X-ray profiles of the prepared membranes are shown in Figure 3. The diffraction patterns of the PU and blends exhibited an amorphous halo peak at 20°, related to the amorphous structure or diffractions from the small crystals [35]. A sharp crystalline peak was observed at 20° for the PVA sample, which also appearred in the PU/PVA200–30 and PU/PVA200–40 samples. However, the intensity of the peak in the MMMs decreased with the silica loadings (Figure 3b). It seems that the presence of the silica in the soft segments disrupted the chain packing and crystallinity. This observation was reported elsewhere [31].

The SEM images of the PU/PVA blends are shown in Figure 4. The miscibility of the blend membranes improved by increasing the PVA200 concentration. The dispersion of the particles is shown in Figure 5. It is clear that the tendency of particles to agglomerate is severe at higher filler loadings (PU/PVA200–40), but they dispersed well at the lower concentration (5 wt %).

### 3.2. Gas Separation Properties

Table 1 presents the influence of PVA200 content on the separation properties of PU/PVA200 blend membranes. The permeability of pure gases through the PU and the PU/PVA200 membranes decreased as follows: P(CO_2_) >> P(CH_4_) > P(O_2_) > P(N_2_). Higher permeability of CO_2_ compared to other gases was attributed to its lower kinetic diameter and higher condensability compared to other gases; therefore, it can make strong interactions with polar groups of C–O–C in the soft segments of PU and OH groups in PVA [37]. Furthermore, higher permeability of CH_4_ compared to the smaller molecules of N_2_ and O_2_ indicates that the solution mechanism plays a dominant role in the gas separation properties of PU/PVA200 blends. This is a typical trend in rubbery polymers [29,38,39]. However, the decrease in CH_4_ permeability by increasing PVA content is larger than the decline in the O_2_ permeability. This behavior is attributed to the more glassy state of the blends where the gas permeability is mostly controlled by the molecular sieving mechanism.

The glassy nature and low intrinsic gas permeability of PVA resulted in a lower gas permeation for the blend membranes. The transport property of the penetrants with larger kinetic diameter was more restricted by increasing the amount of PVA200 into PU/PVA200 blends. However, the CO_2_ gas permeability was less influenced due to its smaller size and the strong interactions with the polar OH group of PVA. So, the selectivity of CO_2_/N_2_ and CO_2_/CH_4_ improved more than non-condensable gases such as O_2_/N_2_ selectivity. Moreover, the higher permeability of PU/PVA200–40 than that of PU/PVA200–30 can be attributed to the surpassing of phase separation (Figure 2) from the glassy nature of PVA.

The CO_2_/N_2_ separation performance of the PU/PVA200 blends was compared to the Robeson’s upper bound (Figure 6a). Based on the Robeson’s upper bound, membranes with both high permeability and selectivity are preferred for a specific separation [40,41]. Hence, the transport properties of the blend membrane of PU/PVA200–10 is more preferred to those of the other blends. The presence of the silica particles in the soft segments (as shown in FTIR and XRD) resulted in a lower gas permeability for the MMMs [42]. The non-porous silica acts as the barrier and increases the tortuosity of the gas diffusion path. As shown in Table 2, the order of gas permeability reduction through the PU/PVA200/silica MMMs was as follows: CH_4_ > N_2_ > O_2_ > CO_2_. The rate of CO_2_ permeability reduction was lower than the other penetrants, which was attributed to its lower kinetic diameter and higher CO_2_ solubility due to the presence of polar OH groups. Furthermore, the higher permeation properties of the PU/PVA200–10/silica MMMs than that of PU/PVA200–40/silica MMMs were attributed to the lower polymer chain mobility of PU/PVA200–40/silica.

Figure 6b compares the separation performance of PU/PVA200/silica MMMs with the Robeson’s upper bound line. As indicated, the CO_2_/N_2_ separation performance of the prepared MMMs was above the Robeson’s upper bound limit in 1991. Nevertheless, PU/PVA200–10 MMM containing 10 wt % silica nanoparticles was predominant to those of the other MMMs.

Table 3 represents the CO_2_/N_2_ (50/50 vol %) and CO_2_/CH_4_ (50/50 vol %) mixed gas separation properties of the pure PU and PU/PVA200–10-S20 membranes at 10 bar and 35 °C. The CO_2_ permeability and the selectivity under the mixed gas condition were lower than the values for the pure gas measurement. The competition of penetrants to pass through the membrane caused the differences in the separation properties of the membranes under the pure and mixed-gas feeds. The decrease in CO_2_ permeability was larger for the CO_2_/CH_4_ gas mixture compared to the CO_2_/N_2_. It seems that the CO_2_ permeation was more influenced by the competition with CH_4_. The CO_2_/N_2_ and CO_2_/CH_4_ mixed gas selectivity were also lower than the pure gas data.

## 4. Conclusions

In this study, a series of PU/PVA blend membranes at different PVA loadings was prepared, and the pure gas permeability of CO_2_, O_2_, N_2_, and CH_4_ was tested at 4 bar and 35 °C. The FTIR results revealed more phase mixing of the PU with increasing amounts of PVA200 into the PU/PVA200 blends. The pure gas permeability of the PU/PVA200 blend membranes decreased with increasing amounts of PVA200 up to 30 wt %. The low gas permeability of glassy PVA was the reason for this reduction. However, the OH groups of the PVA allowed the higher CO_2_/N_2_ and CO_2_/CH_4_ selectivity compared to O_2_/N_2_. The incorporation of silica particles into the PU/PVA200 blends increased the CO_2_/N_2_ selectivity by approximately 40%. Based on the Robeson’s plots, the PU/PVA200–10 membrane showed the best gas separation results among the other samples.

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
