# Peer review of "Influence of Blend Composition and Silica Nanoparticles on the Morphology and Gas Separation Performance of PU/PVA Blend Membranes"

_membranes, 2019, doi:10.3390/membranes9070082_

Round 1
Reviewer 1 Report
This manuscript is well structured and described in a correct way. Nevertheless, the results do not appear of relevant significance, compared to other in Robeson's upper-bound. Therefore, the Authors should better describe - although they improved the performance of their membranes with respect to their previous works - why this research should be of particular interest for a general reader, being - as an example - the CO2/N2 selectivity vs CO2 permeability performance clearly below the Robeson's upper-bound.
Furthermore, the improvements are related only to a part of the permeation performance for the proposed membranes (I mean limited to CO2/N2 separation), while most of the experimental data do not result particularly attracting. Also on this regard, the Authors should better justify why that.
Furthermore, a wider experimental campaign should be carried out, also taking into account the permeation performance in case of gaseous mixtures utilization.
Indeed, the ideal selectivity is useful for describing initially the material characteristics, as not is this case. Then, further data should be added in order to understand what happens when two or more of the used gases are present simultaneously during the permeation tests and how their mutual influence may affect the membrane performance.
A moderate English language check should be done.
Author Response
Response to comments from Reviewer 1.
All major new inserted text is also highlighted in the manuscript itself.
1. Comment: This manuscript is well structured and described in a correct way. Nevertheless, the results do not appear of relevant significance, compared to other in Robeson's upper-bound. Therefore, the Authors should better describe - although they improved the performance of their membranes with respect to their previous works - why this research should be of particular interest for a general reader, being - as an example - the CO2/N2 selectivity vs CO2 permeability performance clearly below the Robeson's upper-bound.
Response: We would like to thank the reviewer for careful reading of this manuscript and providing valuable comments that help us to improve the quality of this paper. Also, we greatly appreciate the positive feedback from the reviewer. Although the result is not promising compared to our previous publications, this work tries to study the synergetic effect of silica particles and PVA on the gas separation of PU membranes, which is the first ever report in the polyurethane gas separation membrane. Besides, the CO2/N2 selectivity for the best composition, PU/PVA200-10-S20, is 45, which is around twice that of the pure PU membrane. We also think that this combination can give a clue to the readership to improve the separation performance of rubbery polymer membranes such as Pebax or PDMS.
2. Comment: Furthermore, the improvements are related only to a part of the permeation performance for the proposed membranes (I mean limited to CO2/N2 separation), while most of the experimental data do not result particularly attracting. Also on this regard, the Authors should better justify why that.
Response: Polyurethane membranes are well-known for CO2/N2 and CO2/H2 separations. Besides, they have some reports on the hydrocarbon separation (C3/C1 and C4/C1). Regarding this, the following sentence was added to page 2 for clarity: “Rubbery polymers are well-known materials for the development of membranes with high CO2/light gas selectivity. Moreover, increasing the CO2 solubility is highly demanding for improvement the separation performance, especially for CO2/N2 industries, where the molecular-sieving doesn’t have a significant role. Among rubbery polymers, PUs have been studied for CO2/N2 separation in recent years because of their structure versatility and good filmability. Besides, the ethereal groups in PUs are suitable sites for CO2 sorption.”
3. Comment: Furthermore, a wider experimental campaign should be carried out, also taking into account the permeation performance in case of gaseous mixtures utilization.
Indeed, the ideal selectivity is useful for describing initially the material characteristics, as not is this case. Then, further data should be added in order to understand what happens when two or more of the used gases are present simultaneously during the permeation tests and how their mutual influence may affect the membrane performance.
Response: We agree with the reviewer’s comment that the mixed gas data should be added to have a better understanding of the membrane separation properties under realistic condition. Therefore, the mixed gas (CO2/N2 and CO2/CH4) permeability test was performed on the polyurethane membranes, and the results are shown in Table 3. Also, we added new paragraphs about the obtained result on page 5 and 8:
“The mixed gas permeability of the membranes was measured at 4 bar and 25°C under the gas mixture of CO2/N2 and CO2/CH4 (50/50 vol.%). The composition of the permeance flow was detected by the gas chromatography machine (GC-2014, Shimadzu). The separation factor was determined by equation 3:”
(3)
Where y and x are the mole fraction of each component in the permeate and feed flow, respectively. The mixed gas permeability was calculated as follow:
(4)
“Table 3 represents the CO2/N2 (50/50 vol.%) and CO2/CH4 (50/50 vol.%) mixed gas separation properties of the pure PU and PU/PVA200-10-S20 membranes at 4 bar and 25°C. The CO2 permeability and the selectivity under the mixed gas condition are lower than the values for the pure gas measurement. The competition of penetrants to pass through the membrane causes the differences in the separation properties of the membranes under the pure and mixed gas feeds. The decrease in CO2 permeability is larger for the CO2/CH4 gas mixture compared to the CO2/N2. It seems that the CO2 permeation is more influenced by the competition with CH4. The CO2/N2 and CO2/CH4 mixed gas selectivity are also lower than the pure gas data.“
4. Comment: A moderate English language check should be done.
Response: Thanks for the comment. The author tried to improve the English grammar of the text.

Reviewer 2 Report
(1) The FTIR spectra of polymer blends and MMMs are hard to read. For example, the authors claim “The peaks at 1105 cm-1 and 1112 cm-1 correspond to the anti-symmetric stretching vibrations of C-O-C and the alkoxy groups of urethane”. I couldn’t tell peaks at 1105 & 1112 cm^-1, only see a big peak at around 1100 cm^-1. Considering the authors also discussed peaks at 1105, 1112, 1443, 1640, 1715 & 1720 cm^-1, zoom-in spectra at the range of 1000 -1750 cm^-1 should be provided with proper annotations.
(2) The description of polymer synthesis (section 2.2) is not clear. If the method was described elsewhere, cite references; the reaction scheme should be provided as well. 3 drops of DBTDL, how much? To achieve NCO/OH ratio of 1, the authors added 10 mmol diamines (BDA), but not alcohols, in the mixture of 15 mmol diisocyanates(IPDI) and 5 mmol PTMG; it seems the authors couldn’t achieve NCO/OH=1.
(3) Additionally, basic polymer characterization like gel permeation chromatography (GPC) is needed to provide the molecular weight of the polymer products.
(4) Tune the description “the presence of the silica in the soft segments disrupted the chain packing and crystallinity” for Fig 2b. The peak intensity or the sharpness for the PU/PVA200-40 and MMMs look the same.
(5) Uncertainty of gas permeation results in Table 1 & 2 should be provided.
(6) In Table 1, why the performance of PU/PVA200-40 is off the trend? It has higher permeability and lower gas selectivity than PU/PVA200-30.
Author Response
Response (in black) to comments (in blue) from Reviewer 2
All major new inserted text is also highlighted in the manuscript.
Comment: The FTIR spectra of polymer blends and MMMs are hard to read. For example, the authors claim “The peaks at 1105 cm-1 and 1112 cm-1 correspond to the anti-symmetric stretching vibrations of C-O-C and the alkoxy groups of urethane”. I couldn’t tell peaks at 1105 & 1112 cm^-1, only see a big peak at around 1100 cm^-1. Considering the authors also discussed peaks at 1105, 1112, 1443, 1640, 1715 & 1720 cm^-1, zoom-in spectra at the range of 1000 -1750 cm^-1 should be provided with proper annotations.
Response: The FTIR plot at the range of 1000 -1800 cm-1 was zoomed in as follows:
Fig. 2. FT-IR spectra of PU, PVA200 and PU/PVA200 blends.
Comment: The description of polymer synthesis (section 2.2) is not clear. If the method was described elsewhere, cite references; the reaction scheme should be provided as well. 3 drops of DBTDL, how much? To achieve NCO/OH ratio of 1, the authors added 10 mmol diamines (BDA), but not alcohols, in the mixture of 15 mmol diisocyanates(IPDI) and 5 mmol PTMG; it seems the authors couldn’t achieve NCO/OH=1.
Response: We agree with the reviewer. The synthesis section was rewritten for clarity and the schematic representation of polyurethane synthesis was added to the manuscript:
The PU was synthesized by a two-step bulk polymerization method, described elsewhere [22]. First, an excess amount of IPDI (3.3 g, 15 mmol) was added dropwise to 10 g PTMG (5 mmol) at 70°C under N2 atmosphere, followed by adding three drops (about 0.15 ml) DBTDL as the catalyst. After mechanical stirring for 2h, the exact amount of BDA (0.88 g,10 mmol) was added to the reaction to carry out the chain extension. Scheme 1 shows the PU synthesis steps. The synthesized PU was precipitated and washed with EtOH/water mixture (50/50 vol. %) to remove the residual monomers, and dried under vacuum at 80°C for further use. The average molecular weight (Mw) and polydispersity index of the PU was determined around 65.1 KDa and 1.4, respectively, by gel permeation chromatography (GPC, Shimadzu, 800 series).
Scheme 1. Schemitic illustration of polyurethane synthesis
Comment: Additionally, basic polymer characterization like gel permeation chromatography (GPC) is needed to provide the molecular weight of the polymer products.
Response: The molecular weight of the synthesized polymer was measured by GPC. The related sentence was added to page 3: " The average molecular weight (Mw) and polydispersity index of the PU was determined around 65.1 KDa and 1.4, respectively, by gel permeation chromatography (GPC, Shimadzu, 800 series)."
Comment: Tune the description “the presence of the silica in the soft segments disrupted the chain packing and crystallinity” for Fig 2b. The peak intensity or the sharpness for the PU/PVA200-40 and MMMs look the same.
Response: As clearly shown in the following figure, the intensity of the hydrogen-bonded (free) carbonyl groups decreases (increases) after incorporation of the silica nanoparticles into the blends. We zoomed in the figure in the range of 1600-1800 cm-1 for more clarity.
Fig. 2. FT-IR spectra of silica particles, PU/PVA200 and PU/PVA200/silica MMMs.
Comment: Uncertainty of gas permeation results in Table 1 & 2 should be provided.
Response: The uncertainty of gas transport properties of the prepared membranes has been presented in Tables 1 & 2 as follows:
Table 1. Separation properties of CO2, O2, N2 and CH4 through PU/PVA200 blends.
Membrane | Permeability (Barrer) | Selectivity | |||||
CO2 | O2 | N2 | CH4 | CO2/N2 | CO2/CH4 | O2/N2 | |
PU | 142.0 ± 7.0 | 15.3 ± 0.8 | 5.5 ± 0.3 | 20.1 ± 1 | 25.8 ± 1.3 | 7.1 ± 0.4 | 2.8 ± 0.1 |
PU/PVA200–2.5 | 126.0 ± 6.2 | 12.5 ± 0.6 | 4.5 ± 0.2 | 16.5 ± 0.9 | 28.0 ± 1.4 | 7.6 ± 0.3 | 2.8 ± 0.1 |
PU/PVA200–5 | 102.0 ± 5.1 | 10.1 ± 0.5 | 3.5 ± 01 | 12.7 ± 0.6 | 29.1 ± 1.5 | 8.0 ± 0.4 | 2.9 ± 0.1 |
PU/PVA200–10 | 93.2 ± 4.5 | 8.3 ± 0.4 | 2.9 ± 0.1 | 9.8 ± 0.5 | 32.6 ± 1.7 | 9.5 ± 0.5 | 2.9 ± 0.1 |
PU/PVA200–20 | 68.7 ± 3.5 | 7.8 ± 0.3 | 2.1 ± 0.1 | 6.2 ± 0.3 | 33.4 ± 1.6 | 11.1 ± 0.6 | 3.8 ± 0.2 |
PU/PVA200–30 | 21.6 ± 1.0 | 2.3 ± 0.1 | 0.6 ± 0.0 | 1.9 ± 0.1 | 35.4 ± 1.8 | 11.2 ± 0.6 | 3.8 ± 0.2 |
PU/PVA200–40 | 40.5 ± 2.0 | 4.3 ± 0.2 | 1.3 ± 0.0 | 7.4 ± 0.3 | 31.4 ± 1.5 | 8.8 ± 0.5 | 3.0 ± 0.1 |
Table 2. Separation properties of CO2, O2, N2 and CH4 through PU/PVA200/silica MMMs.
Membrane | Silica wt % | Permeability (Barrer) | Selectivity | |||||
CO2 | O2 | N2 | CH4 | CO2/N2 | CO2/CH4 | O2/N2 | ||
PU/PVA200–10 | 0 | 93.2 ± 4.5 | 8.3 ± 0.4 | 2.9 ± 0.1 | 9.8 ± 0.5 | 32.6 ± 1.5 | 9.5 ± 0.5 | 2.9 ± 0.1 |
PU/PVA200–10-S2.5 | 2.5 | 79.9 ± 4.0 | 6.8 ± 0.3 | 2.3 ± 0.1 | 6.9 ± 0.3 | 34.7 ± 1.6 | 11.6 ± 0.6 | 2.9 ± 0.1 |
PU/PVA200–10-S5 | 5 | 69.1 ± 3.5 | 5.8 ± 0.3 | 1.8 ± 0.1 | 5.4 ± 0.2 | 37.8 ± 1.7 | 12.8 ± 0.6 | 3.2 ± 0.1 |
PU/PVA200–10-S10 | 10 | 58.8 ± 3.0 | 4.9 ± 0.2 | 1.4 ± 0.0 | 4.2 ± 0.2 | 42.9 ± 2 | 14.1 ± 0.7 | 3.6 ± 0.2 |
PU/PVA200–10-S20 | 20 | 38.3 ± 1.9 | 3.4 ± 0.1 | 0.85 ± 0.0 | 2.5 ± 0.1 | 45.1 ± 2.1 | 15.2 ± 0.7 | 4.0 ± 0.2 |
PU/PVA200–40 | 0 | 40.5 ± 2.0 | 3.9 ± 0.2 | 1.3 ± 0.0 | 4.7 ± 0.2 | 31.4 ± 1.6 | 8.6 ± 0.4 | 3.0 ± 0.1 |
PU/PVA200–40-S2.5 | 2.5 | 37.8 ± 1.9 | 3.5 ± 0.1 | 1.1 ± 0.0 | 3.7 ± 0.1 | 33.8 ± 1.7 | 10.3 ± 0.5 | 3.1 ± 0.1 |
PU/PVA200–40-S5 | 5 | 29.1 ± 1.5 | 2.4 ± 0.1 | 0.7 ± 0.0 | 2.7 ± 0.1 | 33.8 ± 1.7 | 11.0 ± 0.6 | 3.2 ± 0.1 |
PU/PVA200–40-S10 | 10 | 24.1 ± 1.3 | 1.9 ± 0.1 | 0.6 ± 0.0 | 2.1 ± 0.1 | 42.9 ± 2.1 | 11.5 ± 0.6 | 3.3 ± 0.1 |
PU/PVA200–40-S20 | 20 | 15.9 ± 0.8 | 1.3 ± 0.0 | 0.4 ± 0.0 | 1.1 ± 0.0 | 45.4 ± 2.2 | 14.0 ± 0.7 | 3.7 ± 0.2 |
Comment: In Table 1, why the performance of PU/PVA200-40 is off the trend? It has higher permeability and lower gas selectivity than PU/PVA200-30.
Response: Fig. 2 reveals that the intensity of the hydrogen-bonded (free) carbonyl groups decreases (increases) by increasing the amount of PVA200 into the PU. As a consequence, the tendency of PVA to interact with the carbonyl groups disrupt the intermolecular hydrogen bonding in the PU structure. Therefore, more explanation was added on page 7, "higher permeability of PU/PVA200-40 than that of PU/PVA200-30 can be attributed to surpassing of phase separation (Fig. 2) from the glassy nature of PVA."

Round 2
Reviewer 1 Report
The manuscript was significantly improved. Therefore, I suggest to accept it.
Reviewer 2 Report
The authors have well addressed my comments. This manuscript is good to go.